# Detection of Salmonella Enterica in Egg Yolk by PCR on a Microfluidic Disc Device Using Immunomagnetic Beads

**DOI:** 10.3390/s20041060

**Published:** 2020-02-15

**Authors:** Izumi Kubo, Mitsutoshi Kajiya, Narumi Aramaki, Shunsuke Furutani

**Affiliations:** 1Graduate School of Engineering, Soka University, 1-236 Tangi, Hachioji, Tokyo 192-8577, Japantmi_25v17_sle@docomo.ne.jp (N.A.); 2Biomedical Research Institute, National Institute of Advanced Industrial Science and Technology (AIST), 1-8-31 Midorigaoka, Ikeda, Osaka 563-8577, Japan; shunsuke-furutani@aist.go.jp; 3Advanced Photonics and Biosensing Open Innovation Laboratory, National Institute of Advanced Industrial Science and Technology (AIST), 2-1 Yamadaoka, Suita, Osaka 565-0871, Japan

**Keywords:** immunomagnetic beads, Salmonella *enterica*, PCR, invA, egg yolk

## Abstract

Salmonella *enterica* is a pathogenic bacterium that causes foodborne illness. One of the vehicle foods of *S. enterica* are chicken eggs. Efficient collection of the bacterium is necessary to detect it specifically. We developed a method to detect *S. enterica* by PCR on a microfluidic disc device using a fluorescent probe. Salmonella *enterica* cells were isolated in the microchambers on the device, followed by thermal lysis and PCR targeting with the invA gene, a gene specific to *S. enterica*, were observed by measurement of the fluorescent signal that resulted from gene amplification. However, the developed method was unable to discriminate viable cells from dead cells. Consequently, in this study, magnetic beads modified with anti-Salmonella antibody were utilized to detect viable Salmonella cells from egg yolk prior to PCR on the device. While using the antibody-modified beads, egg yolk components, which inhibit PCR, were removed. The collected cells were subsequently detected by PCR of the invA gene on a microfluidic disc device. This method enabled the detection of viable cells without the inhibition of PCR by any egg component. *S. enterica* was detected at 5.0×10^4^ cells mL^−1^ or at a higher concentration of egg yolk within 6 h including the sampling time.

## 1. Introduction

Salmonella *enterica* is a dangerous pathogen that causes outbreaks of foodborne illness [1]. The vehicle foods of *S. enterica* are eggs, meat, and foods made of eggs or meat [2,3,4,5]. Cells of *S. enterica* exist in the intestines of chickens as a coliform bacterium, allowing the easy contamination of chicken meat and eggs [6,7]. Infection of *S. enterica* can occur not only from contaminated eggs or meat but also from its contamination through food processing, transportation, packaging, sales, cooking, and serving. When contaminated food is consumed, after an incubation period of 6 to 48 h, symptoms such as vomiting, diarrhea, and/or fever appear [8,9].

In order to prevent outbreaks, rapid detection of the origin of this bacterium in food is necessary. A conventional method to determine the origin of a bacterium requires multiple culture steps. Coupled with limited dilution analysis, a conventional culture-based method is highly sensitive, and few colony forming units per mL (CFU/mL) can be detected. However, it takes as long as 5 to 6 days to know whether food is contaminated by *S. enterica* [10].

On the other hand, the polymerase chain reaction (PCR) has recently been used for the rapid detection of pathogenic bacteria. The PCR can amplify specific genes of *S. enterica* to millions of copies within 2 h [11]. There are two major methods to detect a PCR amplicon. Electrophoresis is a conventional and inexpensive method, but after PCR, it takes one hour to separate and detect the PCR amplicon [12]. A method that does not need electrophoresis after PCR, such as real-time PCR, would enable rapid detection of a gene. Real-time PCR is based on the binding of a fluorescent probe to the amplicon allowing fluorescent measurement to provide information about the increase of the amplicon. However, reagents used for real-time PCR are expensive, so a reduction in the amount of reagents used is desirable to inspect many samples [13,14].

We developed an original microfluidic device with a compact disc-liked shape for cell isolation [15]. This device has 24 microchannels that spin from the center to the periphery, and each microchannel has 300 or more microchambers. The volume of each microchamber is approximately 1 nL [16]. Detection of bacterial cells on the device was proposed and performed based on PCR on the device.

In this system, at first, bacterial cells are suspended in PCR reagent which is used to detect the specific gene of the target bacterial cells. When 1 μL of a liquid suspension of the microbial cells including the target bacterial cells is introduced into an inlet of a microchannel, centrifugal force can isolate these cells in each microchamber during rotation of the device [15]. After cells are entrapped in the microchambers, cells are lysed in the chambers by heat treatment at 95 °C. Without taking out cells or genes from the cells after lysis, PCR of a specific gene of the entrapped cells can be performed on the device. This process is referred to as hot cell-direct PCR [17]. Using a fluorescent probe for the PCR product, a PCR amplicon can be detected under an epifluorescent microscope [18]. Throughout the process, an entrapped cell can be detected with the use of this device. Use of a fluorescent probe enables a reduction in analysis time compared with the electrophoretic detection method. Moreover, microfluidic detection reduces the consumption of PCR reagents which are used in real-time PCR detection.

In our previous study, we developed a method to detect *S. enterica* by PCR on this microfluidic disc device. *S. enterica* cells were isolated in the microchambers on the device, followed by thermal lysis and PCR targeting of the *invA* gene, which is a specific gene of *S. enterica* [19]. It was confirmed that *S. enterica* at a concentration of 50,000 cells/mL could be detected with the PCR reagents, smaller than one-tenth of Real-time PCR.

Furthermore, we detected *S. enterica* in chicken meat using this device. First, *S. enterica* was separated from chicken meat in a stomacher bag (Atect, Osaka, Japan) and collected by centrifugation using Percoll^®^. *Salmonella enterica* cells collected at 460,000 CFU/g could be detected within 4 h including sampling and detection [11]. Although the *invA* gene from *S. enterica* could be detected efficiently using this procedure, the gene from viable cells might not be distinguished from the gene in dead cells. Viable *S. enterica* cells are known to cause critical illness, but dead cells do not cause illness [20]. Detection of viable pathogenic cells is necessary to prevent an outbreak. For this reason, Ohtsuka and coworkers [5] performed an enrichment culture for 20 h before amplification of the *S. enterica* gene to detect viable cells.

Eggs are another main vehicle of *S. enterica.* The bacterium contaminates raw eggs that are used widely to produce mayonnaise and salad dressing. Since raw eggs form a very thick liquid that contains proteins and lipid, it is not easy to collect bacterial cells from raw eggs. During sample preparation, Salmonella cells are collected by filtration, centrifugation, and the use of immunomagnetic beads on which an anti-*Salmonella* antibody is modified.

The aim of this study was to establish a method to collect viable bacteria removing the egg yolk components from *S. enterica* in an egg sample and to apply hot cell-direct PCR on the microfluidic disc device to detect the *invA* gene. The established method can offer a rapid and economic formula to detect viable *S. enterica* in eggs.

## 2. Materials and Methods

### 2.1. Bacterial Cells and Medium

*S. enterica* was purchased from RIKEN BRC (Tsukuba, Japan). Nutrient broth (Sanko Junyaku Co., Ltd., Tokyo, Japan) with 0.5% NaCl medium (NB medium) (pH 7.2) was used as the culture medium for *S. enterica* cells. *S. enterica* cells were cultured overnight in the liquid media at 37 °C. Cell concentration was confirmed under an optical microscope (BH-2, Olympus, Tokyo, Japan) and diluted to the desired concentration with 10 mM phosphate buffer saline solution (PBS). To obtain dead cells, the cultured cells were autoclaved at 120 °C for 20 min. Buffered peptone water (BPW) (pH 7.2) was used to dilute egg yolk.

### 2.2. Collection of Cells Using Percoll^®^

Bacterial samples from egg yolk as the food sample were prepared as follows. Sterile egg yolk was obtained from Atect (Osaka, Japan) as the food sample. Five mL of egg yolk and 45 mL of BPW were mixed to form an egg yolk solution and one mL of *S. enterica* suspension was spiked to the egg yolk solution. *S. enterica* cells were collected from a cell suspension in egg yolk solution by centrifugation, performed as follows. Egg yolk solution (1 mL) was spiked by the cells to 5 × 10^7^ cells/mL and mixed with 0.7 mL of Percoll^®^ (GE Healthcare Japan Corp., Tokyo, Japan) which is a low viscosity gradient medium for bacterial cell preparation [21]. After the mixture was centrifuged at 14,000 rpm for 15 min at 4 °C, 0.5 mL of precipitate was collected as the *S. enterica* sample. After washing the sample with 10 mM PBS three times, 5 μL of the obtained precipitate was mixed with 20 μL of PCR reagent and used for hot cell-direct PCR.

### 2.3. Filtering S. Enterica from Egg Yolk

Polycarbonate membrane filters (ISOPORE^TM^ membrane filters, MerckMillipore, Tokyo, Japan) with a pore size of 3.0, 0.8, and 0.4 μm was examined for the filtration of *S. enterica* cells. Five microliters of sample spiked by *S. enterica* cells (5.0 × 10^5^ CFU/mL) were directly applied to a filter and filtered cells were counted under an optical microscope.

### 2.4. Collection of S. Enterica from Egg Yolk Using Immunomagnetic Beads

To collect *S. enterica* cells from egg yolk, the use of immunomagnetic beads was investigated. The method to collect viable *S. enterica* cells from egg yolk using immunomagnetic beads (Dynabeads^®^ anti-*Salmonella*, Applied Biosystems by Life Technologies, Tokyo, Japan) is described next (Figure 1). The *S. enterica* cells were spiked to a sterile egg yolk, and one volume of the egg yolk sample was mixed with four volumes of BPW to lower the viscosity of the sample according to a suggested method [22]. The spiked concentration added to one mL of egg yolk was 5 × 10^3^–5 × 10^7^ cells/mL. In a solution with low viscosity, bacterial cells are expected to bind efficiently to the antibody on the beads. At first, 1 mL of the sample solution and 10 μL of the bead suspension (5 × 10^8^ beads/mL) were mixed in a micro tube (1.5 mL, CF-0150, INA-OPTICA, Osaka, Japan). In preliminary experiments, the volume of the bead suspension was examined in 10–40 μL, and 10 μL of the suspension was enough to bind 10^8^ cells. The sample solution and beads suspension were incubated for 20 min at room temperature to allow bacterial cells to bind to the antibody on the beads. Thereafter, beads were collected by placing the tube in a magnetic plate (DynaMag-2^®^, Applied Biosystems by Life Technologies) for 3 min and washed four times with PBST (0.15 M NaCl, 0.01 M phosphate buffer, 0.05% Tween-20, pH 7.4).

Washing removes the egg yolk completely, and *S. enterica* cells are trapped on the immunomagnetic beads which can then be collected. Finally, all the collected beads were suspended in 100 μL of PBST.

### 2.5. Colony Formation after Collection of Immunomagnetic Beads

To examine the collection of viable cells by beads, colony formation was performed as follows. One milliliter of the suspension of *S. enterica* in egg yolk was mixed with 10 μL of beads suspension (5 × 10^6^ beads) in a microtube and incubated for 20 min at room temperature as shown in Figure 1. Then, the beads were collected by placing the tube in a magnetic plate as explained in Section 2.4. It was washed with PBST three times and suspended in 1 mL of PBST. After 1000 fold dilution by PBST, 100 μL of the suspension was used for colony formation. As a control experiment, the same suspension of *S. enterica* served for colony formation after the same dilution. The number of colonies was confirmed by culture on NB agar medium at 37 °C for 24 h.

### 2.6. PCR Reagents and Detection of PCR Product in a Microtube

The PCR reagent used was a Cycleave PCR Salmonella detection Kit Ver. 2.0 (TaKaRa, Tokyo, Japan) which is designed to detect the *invA* gene of *S. enterica*. This kit contains not only all the reagents to detect the *invA* gene using a fluorescent probe labeled with 6-carboxyfluorescein (FAM) but also an internal control DNA sequence, primers, and a probe labeled with X-Rhodamine (ROX) to confirm PCR performance. The internal control DNA works by investigating whether there are interference compounds in the sample or not. When the internal control DNA shows a positive signal after PCR, this indicates the absence of PCR inhibition while in the same sample, a signal of the target gene is not increased, indicating that the concentration of the target gene in the sample is below the detection limit. When both target and internal control are not detected, this indicates that PCR did not occur properly and that a reaction inhibitory factor exists in the sample [11]. The reaction mixture without *S. enterica* was used as the negative control.

In the PCR in a micro tube, cultured *S. enterica* recovered by Percoll^®^ was used as a template for PCR of the *invA* gene. The PCR was performed with a 7500 Real-Time PCR System (Applied Biosystems, Tokyo, Japan). Thermal cycling was initiated at 95 °C for 2 min to lyse *S. enterica*, followed by 40 cycles of 95 °C for 5 s, 55 °C for 10 s, and 72 °C for 30 s according to a previously reported method [11].

### 2.7. Microfluidic Disc Device

The microfluidic disc device was designed to isolate small particles such as cells and micro beads [15]. The microchannel and microchambers were fabricated on a silicon wafer by deep reactive ion etching based on photolithography according to a previously reported method [19]. The prepared silicon wafer with microchannels and microchambers was bonded to a glass plate with holes for inlets and vents [19]. As shown in Figure 2a, on the microfluidic disc device, zig-zag-shaped microchannels were arranged, and 313 microchambers were arrayed on each microchannel (Figure 2b). The sizes of microchannels and microchambers are shown in Figure 2d. The surface of the microchannels and microchambers was modified by triethoxymethylsilane (Wako, Osaka, Japan) to prevent the migration of the liquid after it was trapped in microchambers [11].

### 2.8. Detection of S. Enterica on the Microfluidic Disc

Detection of *S. enterica* on the microfluidic disc was performed as follows. The collected cell suspension (2 μL) was mixed with PCR reagent (8 μL). A mixture of cell suspension (1 μL) was applied to the inlet of the microfluidic disc and the disc was rotated for 30 sec at 3000 rpm to entrap cells in microchambers (Figure 2 C). Lysis of cells, and PCR was performed by a previously reported procedure [11]. In brief, thermal lysis was performed for 2 min at 95 °C after the entrapment of cells in microchambers and PCR (40 cycles 95 °C for 5 s, 55 °C for 10 s, and 72 °C for 10 s) was conducted on the disc. Before and after PCR, the disc was fixed on the automatically controllable stage and fluorescent intensity of microchambers was observed and acquired through an epifluorescent microscope (Olympus, Tokyo, Japan) equipped with a CCD camera (Clara, Andor, Northern Island, UK).

Images of microchambers were obtained except for those near the inlet and vent because of limitations in the movable area of the stage. From the obtained fluorescent image, the fluorescent intensity of each microchamber was measured. The relative fluorescence intensity (RFI) is expressed as:(1)$$\mathrm{RFI}=\frac{Fluorescence\;intensity\;after\;PCR}{Fluorescence\;intensity\;before\;PCR}\;$$
where RFI indicates the existence of a PCR product.

## 3. Results

### 3.1. Effect of Egg Yolk on PCR

In our previous study, *S. enterica* cells were separated from chicken meat with the use of a stomacher bag with filter which is useful for separating small bacterial cells from solid samples, such as meat, collected by centrifugation using Percoll^®^ [11]. The egg sample was a liquid, not a solid. It was considered that the stomacher bag was not effective to separate bacterial cells from an egg sample, so centrifugation using Percoll^®^ was examined in this study to collect *S. enterica* cells. We investigated first whether the collection of *S. enterica* cells using Percoll^®^ could effectively remove egg yolk components or not. To assess the effectiveness, the collected samples in the tubes were examined by PCR as described in Section 2.6.

Without egg yolk, precipitate from *S. enterica* suspension in BPW after centrifugation showed an increase in FAM PCR signals from the *invA* gene to a 2.5 fold higher rate of fluorescence relative to before PCR while ROX PCR signals from the internal control gene also increased by 10.4 fold (Figure 3). These ratios of fluorescence after PCR were almost the same as those after collection of *S. enterica* cells from meat samples [11]. This means that without egg yolk, PCR was performed without inhibition. However, the precipitate from the egg yolk sample with and without *S. enterica* cells showed little increase of both FAM (*invA*) and ROX (internal control) PCR signals. This implies that egg yolk components inhibit PCR. This result indicates that the egg yolk component(s) remaining in the precipitate after the centrifugation using Percoll^®^ inhibited PCR. The sample, when diluted 10 times by BPW, showed a small increase in PCR signals, whereas a 160 fold dilution of BPW showed almost the same increase in PCR signals as when no egg yolk was present. A 10 fold dilution was not enough to avoid inhibition of PCR by egg yolk components. Although a 160 fold dilution was enough to avoid the inhibition, the concentration of *S. enterica* would be lower than the detection limit following this dilution.

To perform PCR without inhibition, it was clear that in the process of collection of *S. enterica* cells, the egg yolk component should be completely removed, but centrifugation using Percoll^®^ was not an effective strategy to remove the egg yolk component.

### 3.2. Filtration of Egg Yolk

An attempt was made to separate bacterial cells from the egg yolk component by filtration with the use of a polycarbonate membrane filter with pores that were smaller than bacterial cells with sizes of approximately 1 μm according to our observations under an optical microscope. It was examined whether there was an appropriate pore size to trap *S. enterica* cells and remove egg yolk components. The filter was that was most appropriate to trap bacterial cells was initially examined. As shown in Table 1, a filter with a pore size of 3.0 and 0.8 μm could not trap cells completely, while a filter with a pore size of 0.5 μm trapped 91% of cells (Table 1). Then, the sample suspended in egg yolk solution was applied to the filters. Although the suspension passed through filters with a pore size 3.0 and 0.8 μm, it could not pass through the filter with a pore size of 0.5 μm. Thus, *S. enterica* cells in the egg yolk sample could not be separated from the egg yolk using membrane filters.

### 3.3. Detection of S. Enterica after Removal of Egg Yolk Using Immunomagnetic Beads

Since the use of polycarbonate membranes was ineffective, the collection of *S. enterica* cells from the egg yolk sample was investigated using immunomagnetic beads (φ = 2 μm, Appendix A). Dynabeads^®^ anti-*Salmonella* react with all current *Salmonella* serovars, and the protocol using these beads serves to determine the presence of viable *Salmonella* [23]. To examine whether the beads collect viable cells, collected cells were counted by colony formation. At the same concentration (5 × 10^5^ cells/mL), the sample after beads collection showed that 4.8 ± 0.2 × 10^5^ cells/mL were obtained. This indicates that 96% ± 4% of viable cells were collected using the immunomagnetic beads.

In order to examine whether *S. enterica* collected from egg yolk using these beads could be detected by PCR or not, we performed PCR in a tube. The fluorescence intensity ratio after the 40th cycle relative to the first cycle was also evaluated (Figure 4).

The fluorescence intensity ratio of *S. enterica* cells collected from egg yolk with beads was about two-fold higher than the negative control and almost the same value as *S. enterica* at the same concentration. Therefore, by using immunomagnetic beads, egg yolk components could be removed completely, and PCR was not inhibited. The egg yolk component was sufficiently removed so that *S. enterica* cells could be collected by microbeads, and the *invA* gene in cells could be detected by PCR. Autoclaved *S. enterica* cells as well as viable cells were examined. The collected beads did not show an increase in fluorescent intensity after PCR. Only viable cells were detected after collection by immunomagnetic beads.

### 3.4. Detection of S. Enterica by PCR on The Microfluidic Disc

*Salmonella enterica* cells were successfully collected from egg yolk with the use of the immunomagnetic beads. The *S. enterica* collected from egg yolk using these beads served for the detection of the *invA* gene by PCR on the microfluidic disc. The obtained suspension of beads after binding to bacterial cells and removal of egg yolk was mixed with the PCR reagent. The mixture (1 μL) was injected into the inlet of the microchannel on the disc.

Figure 5 shows the after-to-before ratio of fluorescent intensity (RFI) by PCR of each microchamber on the microchannel. Microchambers were numbered from upstream (inlet) to downstream (outlet); this figure shows the RFI of each chamber from upstream to downstream. In our previous study, RFI of the negative control was lower than the threshold value 1.4 [11]. A chamber with higher RFI than the threshold was determined to be a chamber that trapped *S. enterica* cells. When the concentration of *S. enterica* was 5 × 10^5^ cells/mL, chambers with an RFI value that exceeded the threshold could be confirmed.

The concentration dependence of the increase in fluorescence was examined next. Figure 6 shows the average value of the intensity ratio of observed chambers of each concentration of *S. enterica* from the range of 5 × 10^3^ sssscells/mL to 5 × 10^7^ cells/mL which are the spiked concentrations into 1 mL of egg yolk. The averaged value of RFI increased depending on the concentration of *S. enterica*. The regression line was y = 0.176 ln (x) – 0.483 and *R*^2^ = 0.96.

### 3.5. Dependence of the Number of Microchambers that Exceeded the Threshold on the Concentration of S. enterica Cells

Figure 7 shows the number of chambers that exceeded the threshold RFI of each concentration of *S. enterica*. The chambers that exceeded the threshold could be confirmed in the case of 5.0 × 10^4^ cells/mL or a higher concentration of *S. enterica* cells. Moreover, the number of chambers that exceeded the threshold increased depending on the concentration of *S. enterica*. In addition, the regression line was y = 21.5 (± 0.91) ln(x) – 40.6 (±1.2) and *R*^2^ = 0.99. The microchambers that exceeded the threshold RFI were considered to have entrapped the immunomagnetic beads binding *S. enterica* cell(s). The number of immunomagnetic beads which bind *S. enterica* cell(s) depends on the concentration of cells. The number of microchambers which trap immunomagnetic beads to which *S. enterica* cell(s) are bound must depend on the concentration of cells. Thus, a linear relationship between the number of microchambers that exceeded the threshold and the concentration of cells indicates that the proposed procedure, including collection of bacteria using the beads, entrapment of the beads in the microchambers, and PCR, reflects the actual concentration of *S. enterica* cells.

The collection of *S. enterica* cells using immunomagnetic beads from egg yolk and PCR was also reported by Vinayaka et al. [24]. In comparison with the same sampling of *S. enterica* suspended in buffer solution, egg yolk did not affect PCR and almost the same signal and linearity were observed [24]. In this study, the observed detection limit and detectable range were comparable with the observed result without using egg yolk in our previous study [19].

Therefore, it was shown that *S. enterica* collected from egg yolk using beads at 50,000 cells/mL could be detected by PCR on a disc. The detection limit was consistent with a previously reported value [19]. After the collection of *S. enterica* cells, the detection limit did not change. This indicates that the collection was performed without a significant loss of cells.

Using the proposed method, *S. enterica* cells in egg yolk were detected by PCR on a microfluidic disc within 6 h. Conventional PCR takes almost 2.5 h, including sample preparation, collection of cells from the egg yolk, mixture with the collected beads with PCR reagent, as well as observation and fluorescent measurement of each microchamber under an epifluorescent microscope. It takes 2 h to perform PCR on the device. After PCR, fluorescent measurement of each microchamber and calculation of RFI is carried out. Finally, the RFI of each chamber is evaluated to determine which chamber exceeds the RFI threshold. This step requires less than 1 h. In total, within 6 h, a measurement can be completed.

## 4. Discussion

Centrifugation with Percoll^®^ and filtration through polycarbonate membranes were a potential candidate method to separate viable *S. enterica* cells from an egg yolk sample. In this study, the cells could not be separated by these methods successfully, since egg yolk is very thick, and its density is almost the same as bacterial cells [25]. To separate bacterial cells from food samples with similar density, a centrifugal method was investigated [25]. It needs several centrifugal steps with different speeds and times and density gradient procedures. The obtained bacteria could be detected by real-time PCR. This method needs many steps and takes time. In comparison with filtration and/or centrifugal methods, a sampling method based on immunomagnetic beads can separate *Salmonella* cells from egg yolk component(s) easily and sufficiently. This component(s) tends to inhibit PCR. Using this method, viable cells were detected after PCR of a specific gene, *invA*. The gene was not detected in dead cells. The denaturation of antigen on the surface of cells might prevent them from binding to the immunomagnetic beads.

In the procedure to collect *S. enterica* cells spiked to egg yolk using immunomagnetic beads, the spiked concentration was 5 × 10^3^ − 5 × 10^7^ cells/mL to 1 mL of egg yolk. In order to remove egg yolk, one fifth of the cell suspension (1 × 10^3^ − 1 × 10^7^ cells) was mixed with 10 µL of immunomagnetic beads (5 × 10^6^ beads) to collect cells. The number of beads exceeded the number of cells except at the highest concentration (5 × 10^7^ cells/mL). The size of the beads was 2 µm in diameter (Appendix A) and larger than the size of *S. enterica* cells. Two or more cells can bind to the surface of one bead.

As shown in Figure 6 and Figure 7, a linear relationship between the RFI of PCR and the concentration of cells was observed. This implies that the number of immunobeads was enough to collect the examined concentration of cells. The lowest detectable concentration was 5 × 10^4^ cells/mL, as shown in Figure 7. As for the detection of *S. enterica* from chicken meat, 4.6 × 10^4^ CFU/g was not detectable without cultivation [11]. After the collection of cells by the beads, the volume was reduced to 100 µL (0.1 mL) from 1 mL, as described in Section 2.4. This indicates that a 10 fold concentration was achieved after collection. Sample preparation using beads resulted in a lower concentration of the sample, which was detected without cultivation. The permissible concentration of *S. enterica* in food is not yet demonstrated. If a more sensitive assay is needed, the collected sample needs to be cultivated. However, when using immunomagnetic beads, a concentrated sample can be obtained from egg yolk. This method will reduce the cultivation time and shorten the time needed to detect bacteria, including detection on the microfluidic device, relative to direct cultivation methods from egg yolk samples. This difference will be examined in the near future.

The detection of bacteria is necessary for a sensor device because detection of the existence of bacteria is more important than quantification (how many cells exist in the sample) when sensing pathogenic bacteria. The signal (RFI) of the proposed sensor depended on the concentration of the bacteria. This sensor device has the potential for quantification.

The collection of *Salmonella* cells using immunomagnetic beads and PCR of the *hil A* gene was also recently reported by Vinayaka et al. [24] as a rapid detection method. However, the method coupled gel electrophoresis and real-time PCR using SYBR^®^ green. Our method does not require gel electrophoresis after PCR and is more rapid than their method.

## 5. Conclusions

*S. enterica* was collected from egg yolk using a small amount (10 μL) of immunomagnetic beads. Fifty thousand cells/mL or a higher concentration of *S. enterica* cells in egg yolk was detected by PCR on the microfluidic disc within 6 h, and the PCR reagent used in this method was less than one-tenth of real-time PCR in a micro tube. Since immunomagnetic beads bind viable *Salmonella* cells, this method provides a rapid and economic formula to detect viable cells without the inhibition of PCR by the egg component. This method may contribute to preventing *Salmonella* outbreaks.

## Figures and Tables

**Figure 1 sensors-20-01060-f001:**
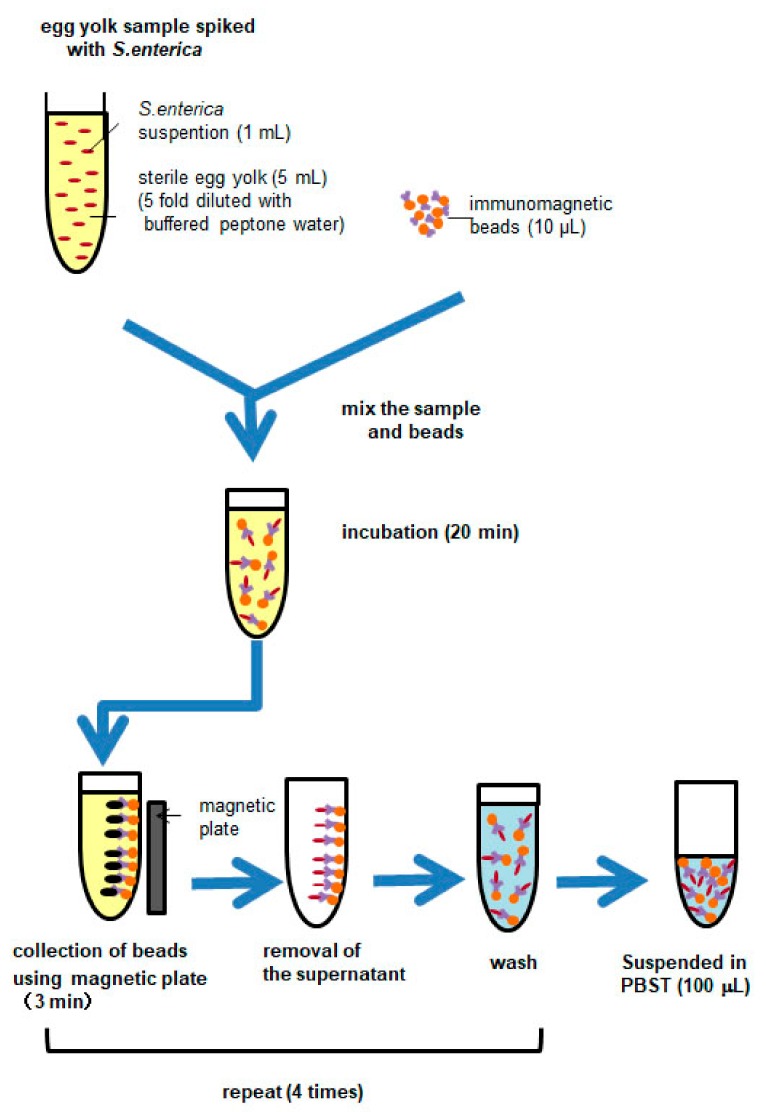
Sample preparation procedure from egg yolk sample spiked with *Salmonella enterica* using immunomagnetic beads: PBST (0.15 M NaCl, 0.01 M phosphate buffer, 0.05% Tween-20, pH 7.4)

**Figure 2 sensors-20-01060-f002:**
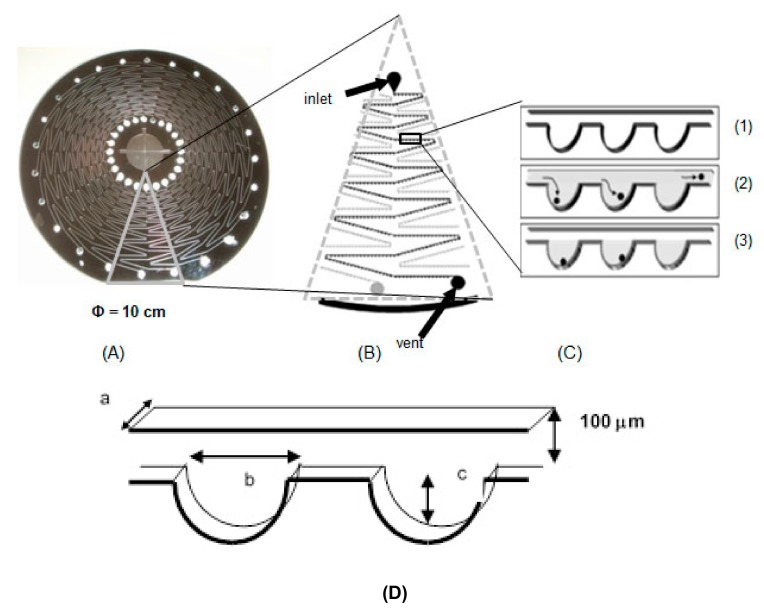
The microfluidic disc device: (**A**) micro fluidic disc device; (**B**) microchannel on the microfluidic disc device; (**C**) microchambers and entrapment of cells. (**1**) Microchambers are arrayed along the microchannel; (**2**) cell suspension flows through the microchannel; (**3**) cells are trapped in microchambers by centrifugal force (3000 rpm, 30 s), (**D**) size of microchannel and microchambers: (**a**) 40 μm, (**b**) 300 μm, (**c**) 200 μm.

**Figure 3 sensors-20-01060-f003:**
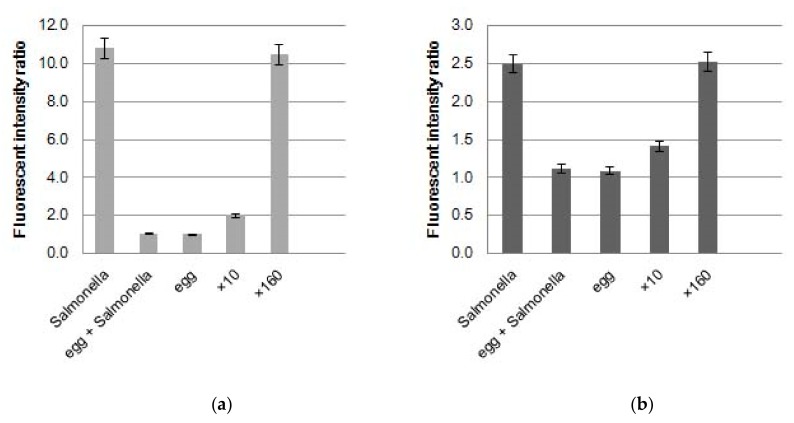
Effect of egg yolk components on fluorescence intensity via PCR: (**a**) PCR signal of 6-carboxyfluorescein (FAM) from the *invA* gene; (**b**) PCR signal of X-Rhodamine (ROX) from the internal control, *N* = 3.

**Figure 4 sensors-20-01060-f004:**
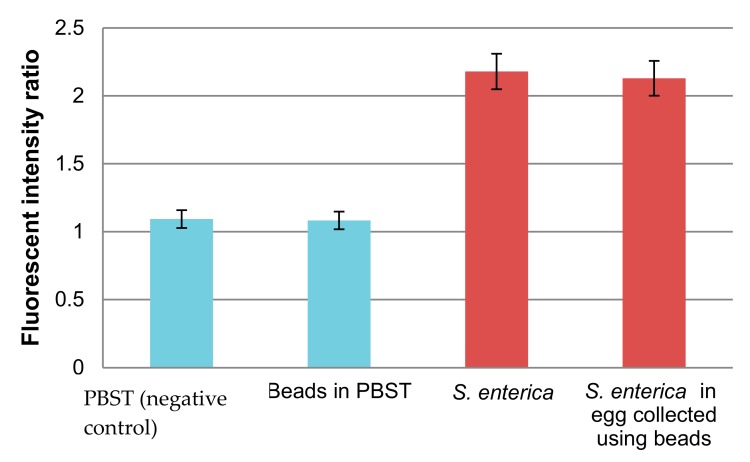
Effect of immunomagnetic beads on the PCR of *invA* in a tube. Concentration of *S. enterica* = 5 × 10^5^ cells/mL, *N* = 3.

**Figure 5 sensors-20-01060-f005:**
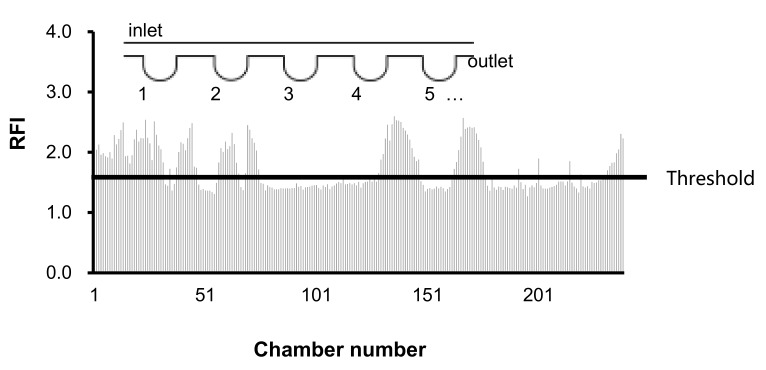
Detection of *S. enterica* by PCR on the microfluidic disc.

**Figure 6 sensors-20-01060-f006:**
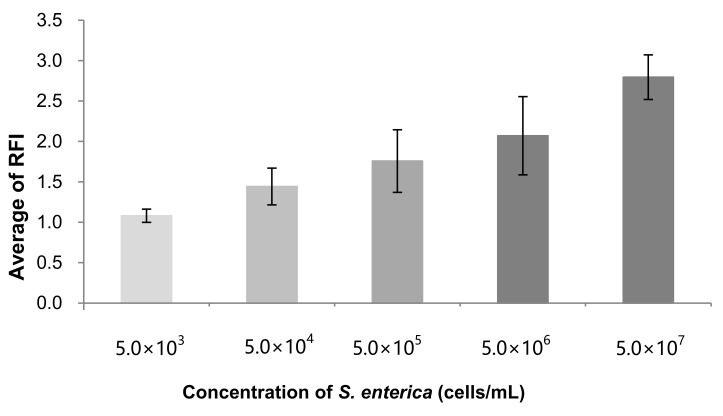
Concentration dependence of *S. enterica* cells on average RFI.

**Figure 7 sensors-20-01060-f007:**
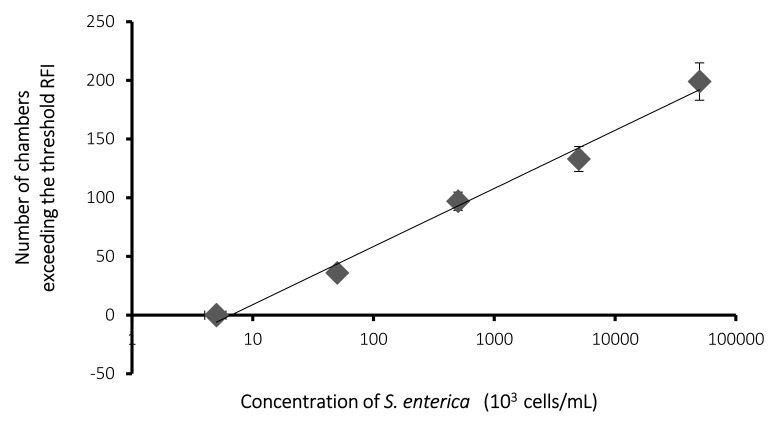
Dependence of the number of microchambers that exceeded the threshold RFI on the concentration of *S. enterica*. *N* = 3.

**Table 1 sensors-20-01060-t001:** Filtration of *Salmonella enterica* in egg yolk solution through polycarbonate membranes.

**Filter**	A	B	C
**Pore size**, μm	3.0	0.8	0.4
**Recovery of Cells**, %	0	7	91
**Passage of Egg Yolk Solution**	○	○	×

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
