# Peer review of "Detection of Salmonella Enterica in Egg Yolk by PCR on a Microfluidic Disc Device Using Immunomagnetic Beads"

_sensors, 2020, doi:10.3390/s20041060_

Round 1

Reviewer 1 Report

  The rapid response to threaten from foodborne pathogens including Salmonella etc. requires reliable rapid detection of these pathogens. The analytical field has been developing new and improved strategies to mitigate this global public health challenge, however, most of the efforts are still laborious, time-consuming and often involve an additional enrichment step for optimal detection, e.g., the incubation steps normally could take up to 48 h. This manuscript and especially its previous direct-related work ( the references therein e.g. 17 [Furutani,S.;Kajiya, M.; Aramaki,N.;Kubo, I.Rapid Detectionof Salmonella enterica in Food Using a Compact Disc-Shaped Device,Micromachines, 2016, 7,10-19.]) could be noted as one of such efforts.   

  Compare with the content in the reference 17, this manuscript looks a small extension to complement some lacks in the former on the detection of living Salmonella enterica. Its main aim is just for clearing about how could be better in getting the bacteria from the egg yolk, through testing three different but common-available separation ways, namely, centrifugation, filter, immunomagnetic beads, for the collection of bacterium prior to performing PCR on their previous developed microfluidic disc device. As these collection methods are very common measures in bioanalysis, there is not only nothing new but also a most probably predictable result before doing such experiments. Except this, the overall workflow in the manuscript is bascially the same as before (the author’s own previous published works). In addition, the shortcomings of this manuscript may also be lack of the comparison with most of current arts for this pathogen detection, and be too more self-citing references, and some cited references are too earlier. Therefore, I think that it is not suitable for publication in this journal.

Author Response

Thank you for your valuable comments.

The manuscript was revised according to the comments of all the reviewers.

Revised part is shown in red in the text.

Some references of current work to be compared to this work were referred in discussion.

Even if the result is predictable, it should be proved by evidence in science. The evidence should be open to public. Authors wish to publish our results as the evidence. Although the collection of Salmonella using immune magnetic beads was proved to be effective, transferring the collected cells on the immune magnetic beads to microchambers by centrifugal force has not been reported. The authors believe that the efficiency of the method should be reported in this manuscript.

We hope that the explanations and revisions of our work are satisfactory, now.

Reviewer 2 Report

Kubo et al. report on the using of immunomagnetic beads for the isolation of S. enterica from egg yolk for their microfluidic PCR detection. Methods such as Percoll® gradient solution and membrane filtration were found to be ineffective in preparation of sample from egg yolk. A detection limit of 5 × 105 cells/mL was demonstrated using the immuno-magnetic isolation method for the sample preparation. Overall, the manuscript is well structured and experiments were appropriately designed. I will recommend it for publication if the following concerns are properly addressed.

Major

Need evidence that immuno-magnetic beads only capture live cells. Authors reported the recovery rate for the filtration method (e.g., 91%), what was the recovery rate of the bacteria using the immuno-magnetic beads? This cannot get from the fluorescent intensity ratio. Is the spiking concentration reasonable when testing the effectiveness of immuno-magnetic bead method? What is the critical concentration or actual dangerous concentration of S. enterica in egg yolk? What was the concentration of the beads? Does increasing bead number help get better recovery? Since the threshold is 1.4, the detection limit of this method is 5 × 105 cells/mL according to Fig 6? Is this sensitive enough in the reality? More discussion for Fig. 7. Is the number of positive chambers linearly correlated to the cell concentration? This Figure would need data points of controls with different concentration, which are the cells spiked in PBS or BPW.

Minor

In the method section 2.2, please include the spiking concentrations of the S. enterica. Also, in the last sentence, how much is the “one part”? How many parts in total? In the method section 2.3, is the 5 mL sample spiked with S. enterica the sample after centrifugation? Please specify the spiking concentrations. In the method section 2.4, what was the spiking concentration for the immunomagnetic beads experiments? In the section of microfluidic PCR, why only the results of 200 chambers rather than 313 chambers are reported? “This means the collection was performed without loss of the cells.” This statement should be changed into “without significant loss of cells” as there is no direct evidence supporting their claim of no cell loss. Have the authors tried to detect S. enterica from actual egg yolk? What are the error bars in Fig. 3, 4, 6 and 7? Is the 6-hour the testing time including all the sample preparation steps using magnetic beads and PCR detection? Moderate English editing is necessary to correct grammar issues and to improve the readability of the manuscript.

Author Response

Thank you for reading our manuscript and valuable comments.

The manuscript was revised according to the comments of all the reviewers.

Revised part is shown in red in the text.

The revised points according to your comments are as follows.

The evidence of the capture of viable cell and the recovery rate was added in L 243-246.

The critical concentration of S. enterica in egg is not known. Not detectable in the sample is desirable in the assay. But detectable concentration depends on the detection method. Conventional method is based on several steps of culture, which lacks quantification of the number of cells. The critical concentration I not reported.  Then the authors do not mention the critical concentration of S. enterica in egg.

All the revised parts were shown in red. Concentration of beads, spiked concentration of S. enterica cells and so on.

The sensitivity of this method was discussed.

The reason why the number of observed chambers was smaller than 313 was explained in L 177-178.

We hope that the explanations and revisions of our work are satisfactory, now.

Reviewer 3 Report

In abstract author should include how / at what extent the studied method has been improved? Why the reader should care about this work? Need references on “detection of pathogenic bacteria. PCR can amplify specific genes of S. enterica to millions of copies within 2 h” Need references on “but after PCR it takes a long time to detect the PCR amplicon” “Use of a fluorescent probe enables a reduction in analysis time compared with the electrophoretic detection method.”- Author need to quantify the reduction time Detail is needed on microfluidic substrate prep. such as what metal used to pattern microchannel design? What process used? Photolithography? A more detail separate section on the fabrication step is required. Did pcr performed on separate microtube or within the microchamber of microfluidic device? Sec 2.7 “Lysis of cells and PCR were performed by a previously reported procedure [17]”- though a reference was cited, a brief detail on it might help reader to understand how pcr performed on microfluidic substrate. Figure 2 needs to be more detail. The dimension of the whole substrate, microchannels, and micro chambers needs to be labeled on the figure. at the end, the figure needs to be of high resolution. I believe, Figure 6 and figure 7 both show linear relationship with the X and Y component. Therefore linear regression analysis should be done on these figures, So that more meaningful results / findings can be concluded. The author should discuss limitations of the current study and future work In conclusion section.

Author Response

Thank you for reading our manuscript and helpful comments.

The manuscript was revised according to the comments of all the reviewers.

Revised part is shown in red in the text.

The revised points according to your comments are as follows.

The references about gel electrophoresis and PCR was added.

The reduction of time is one hour and it was mentioned in the revised manuscript.

Preparation method was added in L158-159.

A brief detail of PCR on the device was added in 2.7.

In Figure 2, details of the size of microchannel and microchambers are added.

Regression analysis of Figure 6 and 7 was done. In figure 7 regression line is shown. However, regression line cannot be shown in Figure 6, because this figure is not scattered plot.

About the future work, assay for lower concentration was mentioned in discussion.

We hope that the explanations and revisions of our work are satisfactory, now.

Reviewer 4 Report

Salmonella enterica is popular pathogenic bacteria present in meat or eggs. Authors developed a method method to detect S. enterica via PCR on microfluidic disc. To remove bacteria from the yolk they have used an immunomagnetic beads. The manuscript can be interesting for the readers of the Sensors, I think it can be accepted after a minor revision.

-could Authors describe with more details immunomagnetic beads used in the experiments? what is the chemical composition, size (maybe also TEM or SEM pictures) and the modification on the surface?

-Authors should also mention more details about preparation of the microfluidic device: what are the geometrical parameters of the channels?

-S. enterica should be written in italic in all manuscript

-Fig. 4-7, the background should be changed to white

Author Response

Thank you for your valuable comments.

The manuscript was revised according to the comments of all the reviewers.

Revised part is shown in red in the text.

The revised points according to your comments are as follows.

Immunomagnetic beads modified with anti-salmonella antibody used in this manuscript were commercialized by the company, applied Biosystems by life technologies. The size of the beads can be observed under optical microscope and Fig. S1 was added. However, the information about surface modification of the beads is not offered by the company. Authors cannot mention the surface modification of the beads.

Geometrical parameters of the microfluidic device were added in Figure 2 D.

Back ground of Fig 4-7 is white. We checked it using several displays, Windows OS and Mac OS. We did not change the background of these figures.

We hope that the explanations and revisions of our work are satisfactory, now.

Reviewer 5 Report

The authors show a disc-based method for detection of pathogenic bacteria from egg which they previously showed for other sources. in the current work the authors added a magnetic based purification step before sample introduction into the pcr chip. Even though in the bead immunosensor category this is rather an improvement of sample prep than a manuscript on bead-based sensor technology. The current manuscript is out of scope for the chosen journal and category.

overall the technology works but the state of the art assays for detection of such bacterial contaminations from food even though more time consuming is missing as control experiments to compare. Also sample in data out would require the bead based workflow to happen on the spinning disc, which is currently outside of the pcr chip. Discussion is very superficial und minimalistic. 

Author Response

Thank you for your valuable comments.

The manuscript was revised according to the comments of all the reviewers.

Revised part is shown in red in the text.

The revised points according to your comments are as follows.

The experiment to be compared to this work was mentioned in discussion.

Detection of S. enterica on micro fluidic disc was revised to explain the detail procedure in 2.7.

Discussion about this research was added in L 344-366.

We hope that the explanations and revisions of our work are satisfactory, now.

Round 2

Reviewer 1 Report

The second version of this manuscript has been made several improvements, but the main scientific meaning of its protocol via IMS remains to be given more rationalization if it could be considered to publish. From its abstract on webpage, the reference[20] seems only to indicate the percentage of Salmonella (3.1%) in the causative pathogens to lead to 105 cases of acute diarrhea patients, but no information about the bacteria’s living or dead. This reference[5] showed LAMP method was better than most other PCRs, esp. the LAMP assay was not inhibited by the constituents of liquid egg. This fact may undervalue the authors’ current complex protocols revealed in their manuscript. In addition, the detection time-limit  ‘within 6 h‘ may have more evidences.

Author Response

Thank you for your valuable comments.

The manuscript was revised according to the comments of all the reviewers.

English of the manuscript was corrected by native and professional proofreading.

Revised parts according to your comments are shown in red in the text and the revised parts according to other reviewers are shown in blue.

The responses to your comments were attached after the revised manuscript.

Reviewer 2 Report

The revised manuscript includes more details about the work. However, the authors are recommended to provide a “point-by-point” response to reviewer’s comments. It is rather confusing if the authors have or have not addressed my previous comments in their current response note.

Some additional concerns

Since colony formation assay was used to check the viability, please provide details about how this was done in the method section. The authors suggested that the desired results would be no S. enterica detected and this level is depending on detection method. The question is if the detection limit of 5000 cells/mL is practically sufficient considering the sensitivity of conventional method is a few CFU/mL? In Fig.6, please specify RFI in the title of y axis. Please provide R2 for the equation in Line 318. The advantages of the proposed method should be highlighted in the introduction. English needs be improved, especially for those new text in red.

Author Response

Thank you for your valuable comments.

The manuscript was revised according to the comments of all the reviewers.

English of the manuscript was corrected by native and professional proofreading.

I apologize inadequate response to your first comments.

Revised parts according to your first comments and second comments are shown in purple and green in the text respectively and the revised parts according to other reviewers are shown in blue.

The responses to your comments were attached after the revised manuscript.

Reviewer 3 Report

Some concerns author still needs to address to make this manuscript publishable

The customary way to response to reviewer questions, is to include a detail answer along with what was changed / added in the manuscript. Author should follow this usual process.  A more detail separate section on the device fabrication step is required. Author just added a single sentence. The existing description do not give enough information to the readers.  Linear regression should have error and respective units within the equation, such as y = (m ± error) x unit + (c ± error) unit. Author needs to update all the equations, so that, the reader would easily back-calculate the x value (concentration value) from any RFI value by using the equation.  Line 318, what author meant by "Such linear relationship indicates that the proposed procedure reflects the concentration of S. enterica cells" ? Its not clear. 

Author Response

(The authors gave the same response as above.)

Reviewer 5 Report

The authors improved the manuscript a bit however language needs strong editing. 

Still it is unclear how this bead-separation of viable cells before the already published per chip is in scope for a sensor journal. Where is the sensor located? What is the sensor? It is definitely not the beadprep...

Author Response

Thank you for your valuable comments.

The manuscript was revised according to the comments of all the reviewers.

English of the manuscript was corrected by native and professional proofreading.

Revised parts according to your comments are shown in green in the text and the revised parts according to other reviewers are shown in blue.

Response to your comment is as follows.

Your comment

Still it is unclear how this bead-separation of viable cells before the already published per chip is in scope for a sensor journal. Where is the sensor located? What is the sensor? It is definitely not the beadprep...

Response

The usability of our sensor and explanation was added. (L 388-391)
